# Disulfidptosis: A Novel Prognostic Criterion and Potential Treatment Strategy for Diffuse Large B-Cell Lymphoma (DLBCL)

**DOI:** 10.3390/ijms25137156

**Published:** 2024-06-28

**Authors:** Yu Wang, Yoshiyuki Tsukamoto, Mitsuo Hori, Hidekatsu Iha

**Affiliations:** 1Department of Microbiology, Faculty of Medicine, Oita University, Yufu 879-5593, Japan; m21d9024@oita-u.ac.jp; 2Department of Molecular Pathology, Faculty of Medicine, Oita University, Yufu 879-5593, Japan; tuka@oita-u.ac.jp; 3Department of Hematology, Ibaraki Prefectural Central Hospital, Kasama 309-1703, Japan; mi-hori@chubyoin.pref.ibaraki.jp; 4Division of Pathophysiology, The Research Center for GLOBAL and LOCAL Infectious Diseases (RCGLID), Oita University, Yufu 879-5503, Japan

**Keywords:** diffuse large B-cell lymphoma (DLBCL), disulfidptosis, disulfidptosis-related genes (DRGs), machine learning, immunity, prognosis, treatment

## Abstract

Diffuse Large B-cell Lymphoma (DLBCL), with its intrinsic genetic and epigenetic heterogeneity, exhibits significantly variable clinical outcomes among patients treated with the current standard regimen. Disulfidptosis, a novel form of regulatory cell death triggered by disulfide stress, is characterized by the collapse of cytoskeleton proteins and F-actin due to intracellular accumulation of disulfides. We investigated the expression variations of disulfidptosis-related genes (DRGs) in DLBCL using two publicly available gene expression datasets. The initial analysis of DRGs in DLBCL (GSE12453) revealed differences in gene expression patterns between various normal B cells and DLBCL. Subsequent analysis (GSE31312) identified DRGs strongly associated with prognostic outcomes, revealing eight characteristic DRGs (*CAPZB*, *DSTN*, *GYS1*, *IQGAP1*, *MYH9*, *NDUFA11*, *NDUFS1*, *OXSM*). Based on these DRGs, DLBCL patients were stratified into three groups, indicating that (1) DRGs can predict prognosis, and (2) DRGs can help identify novel therapeutic candidates. This study underscores the significant role of DRGs in various biological processes within DLBCL. Assessing the risk scores of individual DRGs allows for more precise stratification of prognosis and treatment strategies for DLBCL patients, thereby enhancing the effectiveness of clinical practice.

## 1. Introduction

Diffuse Large B-cell Lymphoma (DLBCL) is a malignancy characterized by the proliferation of large B cells, typically exhibiting a diffuse growth pattern [1,2]. As one of the most prevalent and heterogeneous forms of non-Hodgkin lymphoma, DLBCL can be subdivided based on various criteria, including molecular and clinical features. This leads to distinct biological profiles and variable responses to therapeutic interventions [1,2,3,4]. While survival rates have significantly improved (reaching 50–60%), the heterogeneity of DLBCL contributes to varying clinical outcomes among the patients treated with the current standard treatment regimen, consisting of rituximab combined with traditional chemotherapy of cyclophosphamide, doxorubicin, vincristine, and prednisone (R-CHOP) [5]. As high-throughput technology advances, utilizing gene expression profiles for example, the most established method for subtyping DLBCL categorizes it into activated B-cell like (ABC) and germinal center-B-cell like (GCB) subtypes, along with the unclassifiable (UC) subgroups. These molecular subtypes exhibit distinct prognostic significance, emphasizing the importance of molecular classification in understanding DLBCL heterogeneity and guiding treatment decisions [1,4].

Patients with GCB-DLBCL exhibited significantly higher overall survival compared to those with ABC-DLBCL. Specifically, the 5-year progression-free survival (PFS) rates with R-CHOP were 48% for ABC-DLBCL and 73% for GCB-DLBCL [4,5]. High-risk DLBCL patients, especially those classified as unclassified, typically experience poorer progression-free survival and overall survival rates when compared to other subtypes [6]. The delineation of molecular subtypes highlights how gene expression-based molecular classification of DLBCL identifies previously unrecognized clinically relevant disease subtypes. For instance, Wright et al. identified a set of 27 genes that serve as predictors for diagnosing various clinical subtypes of DLBCL [7]. Similarly, Cai et al. devised an expression-based signature consisting of up to 35 genes for subtype classification and prediction of survival outcomes [8]. These genes serve various biological functions, including involvement in focal adhesion, cell cycle regulation, and cell survival pathways. This underscores the multifaceted nature of DLBCL pathogenesis and its potential therapeutic implications.

An emerging cell death mechanism, ferroptosis, characterized by iron-dependent lipid peroxidation [9], has gained prominence as a potential therapeutic strategy in tumor research [10]. The central mechanism of ferroptosis revolves around the precise regulation of the equilibrium between oxidative damage and antioxidant defense [9]. Recent studies have elucidated that SLC7A11 plays a pivotal role in resisting oxidative stress and suppressing ferroptosis triggered by oxidative damage to cell membranes, which is primarily achieved by maintaining intracellular glutathione levels [10]. At the same time, it is noteworthy that SLC7A11 also mediates an alternative form of programmed cell death, known as disulfidptosis [11]. Disulfide stress, leading to disulfidptosis, arises from an imbalance in intracellular redox homeostasis primarily attributed to the abnormal accumulation of intracellular disulfides [12]. This stress can be highly detrimental to cells, as it induces the formation of disulfide bonds between proteins, particularly affecting the actin cytoskeleton [11,12]. Cancer cells frequently encounter elevated levels of oxidative stress and exhibit heightened demands for cysteine. Since cysteine is prone to rapid oxidation to cystine in the extracellular environment, the concentration of cystine in the extracellular space is typically exceeds than that of cysteine by an order of magnitude [13,14]. Therefore, most cancer cells transport extracellular cystine into cells via the cystine transporter SLC7A11, where it is reduced to cysteine using NADPH [10]. The cysteine then participates in the synthesis of glutathione, enabling cancer cells to reinforce their defenses against oxidative stress [13,15,16]. However, since the solubility of cystine is extremely low, the absence of NADPH can lead to aberrant cystine accumulation in the cytoplasm, triggering disulfide stress and subsequent cytotoxicity [17], which includes the disruption of cytoskeletal proteins, ultimately resulting in disulfidptosis [11].

Disulfidptosis, emerging as a novel form of programmed cell death, offers a promising avenue for cancer therapeutics. Although research in this field is still in its early stages, further investigation is warranted to elucidate its distinct mechanism of action and therapeutic applications [15,18,19]. In this study, we initially examined the expression variations of disulfidptosis-related genes (DRGs) in DLBCL compared to various normal B-cell subsets using the public dataset GSE12453. Additionally, we identified eight characteristic DRGs (*CAPZB*, *DSTN*, *GYS1*, *IQGAP1*, *MYH9*, *NDUFA11*, *NDUFS1*, *OXSM*) strongly associated with survival outcomes, using a dataset of 470 clinical samples from the publicly available dataset GSE31315. After classifying DLBCL into diverse DRG endotypes through unsupervised clustering based on the aforementioned DRGs, enrichment analysis and tumor microenvironment analysis were conducted to explore the reason of different prognosis in DRG-induced subtypes. Then, small molecule compounds with potential to prolong prognosis were also screened for the subgroups with poor prognosis. In summary, this study established a prognostic signature related to disulfidptosis, and shed light on the role of disulfidptosis in specific hematological conditions through the subtyping of DLBCL.

## 2. Results

### 2.1. The Landscape of Disulfidptosis-Related Genes (DRGs) between DLBCL and Normal B Cell Subtypes

We first collated a list of 24 DRGs from the published articles (Table 1, Figure 1A). The regulatory interaction of each DRG and the GO/KEGG function enrichment were exhibited as a protein–protein network (Figure 1B) and a pathway heatmap (Figure 1C), which of note shows that the functions of DRGs broadly divided into two functional areas including mitochondrial respiration and actin cytoskeleton.

The pathway–pathway network and GeneRatio of the top pathway are shown in Appendix A. (The indicated abbreviations of each capital letter in Appendix A are as follows: BP stands for biological process, CC for cellular component, MF for molecular function, and KEGG for Kyoto Encyclopedia of Genes and Genomes). The transcriptome relationships were examined using Spearman correlation (Figure 1D and Appendix A, Appendix A). The co-expression tendency of NDUFS1 and MYH10 in normal B-cell samples was reversed in DLBCL samples, and the negative correlation between DSTN and PDLIM1 in normal B-cell samples was also reversed in DLBCL samples. NUBPL and MYL6 exhibited a distinct inverse association in DLBCL that was not observed in normal B cells. Conversely, INF2 and FLNA displayed high correlation as DRGs in both normal B cells and DLBCLs, suggesting a coordinated function.

Additionally, we examined the differential expression of DRGs in DLBCL compared to normal B-cell subsets (Figure 2). In the supervised comparison between the 24 DRGs in DLBCL and plasma cells, seven genes (*CAPZB*, *DSTN*, *IQGAP1*, *MYH10*, *NDUFA11*, *PDLIM1*, *RPN1*) exhibited statistically significant differential expression (|fold changes|>1) in a statistically significant manner (*p* < 0.05) (Figure 2A,F, Appendix A). CAPZB showed the most up-regulation, while RPN1 exhibited the most down-regulation in DLBCL compared with plasma cells. In the comparison of the 24 DRGs between DLBCL and naive B cells, 10 genes (*CAPZB*, *DSTN*, *INF2*, *IQGAP1*, *MYH10*, *MYL6*, *NDUFA11*, *NUBPL*, *RPN1*, *TLN1*) showed statistically significant differential expression (Figure 2B,F, Appendix A). Among them, MYH10 exhibited a 3.19-fold difference, while IQGAP1 showed a −1.82 difference. In total, 5 out of the 24 DRGs exhibited significant differences in DLBCL compare to memory B-cells, including DSTN, FLNB, INF2, MYH10, and NDUFA11 (Figure 2C,F, Appendix A). Specifically, MYH10 showed a 2.92-fold difference, and FLNB shows a 1.58-fold difference, respectively. The regulation of 24 DRGs in DLBCL compared to centrocytes is mostly moderate (Figure 2D,F, Appendix A). In the comparison between DLBCL and centroblasts, 5 out of 24 DRGs showed a significant difference, including IQGAP1 (−2.08-fold change), NDUFA11 (1.52-fold change), NDUFS1, PDLIM1, RPN1 (Figure 2E,F, Appendix A). Figure 2F, presented as a heatmap, illustrates the profiling of the 24 DRGs’ expression level in DLBCL compared to normal B-cell subsets.

### 2.2. DRGs Were Found to Influence the Prognosis Prediction of DLBCL

To explore the impact of DRGs’ expression levels on prognosis, we conducted an unsupervised consensus clustering analysis for 470 DLBCL samples (GSE31312) using the expression data of 24 DRGs (Figure 3A–C, Appendix A). Three distinct subtypes of DLBCL were identified based on the expression patterns of the 24 DRGs, as depicted in the PCA plot (Figure 3D), with 158 samples in cluster 1, 185 samples in cluster 2, and 127 samples in cluster 3. The heatmap (Figure 3E) illustrates the qualitatively different expression profiles of the 24 DRGs across these three clusters. The Kaplan–Meier analysis of overall survival (Figure 3F) reveals a significant difference in survival probability between cluster 2 and cluster 1/3 after 4 years (*p* < 0.001), indicating the predictive potential of DRGs as molecular markers in DLBCL.

### 2.3. Construction of a DRG-Related Risk Signature

Based on the 24 DRGs, 13 genes were subsequently selected through a univariate Cox regression analysis (*p* < 0.05) as potential prognosis risk factors for patients with DLBCL (Figure 4A). Next, the number of genes was further narrowed down to eight through LASSO regression analysis, followed by multivariate Cox regression. These genes included *CAPZB*, *DSTN*, *GYS1*, *IQGAP1*, *MYH9*, *NDUFA11*, *NDUFS1*, *OXSM* (Figure 4B–D, Table 2), which were then utilized to establish a prognostic model for DLBCL patients. The risk score for each patient in GSE31312 was computed based on the following formula:Risk score = 0.189 × Exp(CAPZB) − 0.178 × Exp(DSTN) + 0.185 × Exp(GYS1) + 0.094 × Exp(IQGAP1) − 0.248 × (MYH9) + 0.1 × Exp(NDFA11) − 0.008 × Exp(NDUFS1) + 0.444 × Exp(OXSM).

The patients were categorized into high-risk and low-risk subgroups based on the median risk score. Kaplan–Meier curves revealed a significant trend toward reduced survival in the high-risk group compared to the low-risk group after 2 years (*p* < 0.001, Figure 4E). The distribution of risk scores and survival times between the high-risk and low-risk groups is depicted in Figure 4F. ROC curves were generated with an AUC value of 0.716 for predicting 5 years, indicating the accuracy of the prognostic risk models (Figure 4G).

To investigate deeper into the DLBCL influenced by the risk signature associated with DRGs, we conducted consensus clustering using the expression matrix of the eight DRGs obtained after the univariate Cox and LASSO regression analysis. Unsupervised hierarchical clustering divided the DLBCL population into three clusters (Figure 5A–C). Subsequent prognosis analysis (Figure 5D) unveiled significant differences in prognosis among three clusters, with a median follow-up at 42 months. Across all cohorts, a patient classified within cluster 3, belonging to the low-risk group, exhibits approximately 1.5-times higher odds of survival at the three-year mark compared to a patient assigned to cluster 2.

Furthermore, this survival advantage escalates substantially to threefold at the five-year mark. The distinct clusters depicted in the heatmap revealed various gene expression responses (Figure 5E). *NDUFA11*, *CAPZB*, *IQGAP1*, *GYS1*, *OXSM*, and *MYH9* showed significantly lower expression levels in cluster 3, associated with a relatively good prognosis, compared to cluster 2 with poor prognosis, while the opposite was observed for DSTN and NDUFS1. The expression of the eight DRGs in cluster 1 falls somewhere between that of cluster 2 and cluster 3. Additionally, FRIEND analysis was employed to evaluate the importance ranking of the eight DRGs (Figure 5F). The genes were arranged in descending order based on their mean similarity to each other. Consequently, the prominence of *NDUFA11* at the top of the hierarchy underscores its pivotal role within this gene set, suggesting its potential significance in cellular functions. The Sankey diagram illustrates the composition of the DLBCL subtypes within each cluster and the corresponding risk score levels (Appendix A).

### 2.4. The Differentially Expressed Genes (DEGs) between Cluster 3 Compared with Cluster 1/2

Given the evident prognostic variability among the three clusters, we designated cluster 3, which showed a relatively favorable prognosis, as the reference standard. Subsequently, we computed the fold changes of cluster 1 and 2 relative to cluster 3 and used the resulting log2 fold changes to rank gene lists, facilitating gene set enrichment analysis (GSEA) for cluster 1 (Figure 6A) and cluster 2 (Figure 6B). Compared to cluster 3, the overall distribution of gene expression changes in cluster 1 and cluster 2 appears skewed or shifted toward cell division, with increased expression of cell cycle-related gene sets (positive scores) and decreased expression of cell–cell junction-related gene sets (negative scores). This observation suggests that clusters with poorer prognoses exhibit intensified cell cycle regulation and enhanced cell migration capacities.

The differentially expressed genes (DEGs) between cluster 1 and cluster 3, cluster 2 and cluster 3, and cluster 1 and cluster 2 are presented in Figure 6C. Between cluster 2 and cluster 3, 652 DEGs were identified, with 145 genes overlapping with the DEGs found in the comparison between cluster 1 and cluster 3. Notably, only one gene, NDUFA11, exhibited significant differential expression between cluster 1 and cluster 2. The protein–protein interaction network of DEGs is illustrated in Figure 6D. Additionally, enriched GO and KEGG pathway analysis for DEGs were further analyzed in Appendix A. The biological process (BP) primarily linked to the DEGs encompasses functions such as myeloid cell differentiation (GO:0030099), protein neddylation (GO:0045116), and nuclear DNA replication (GO:0044786). In terms of molecular function (MF), the DEGs showed significant enrichment in functions such as cadherin binding (GO:0045296), heat shock protein binding (GO:0031072), and transcription coactivator binding (GO:0001223). Notably, the only KEGG pathway enriched in DEGs is ferroptosis (hsa04216), along with the ferric iron binding (GO:0008199) of MF, suggesting a potential dysregulation in the iron metabolism pathway, which may be associated with an unfavorable prognosis.

Figure 6D also showcases the top ten hub genes within the DEGs, highlighted in red. These genes were arranged using the Maximum Clique Centrality (MCC) algorithm, with their significance depicted through a gradient shading from darker to lighter tones. The hub genes, along with their interactions with transcription factors (TFs) and non-coding RNAs (ncRNAs), were illustrated in the hub genes–TF-ncRNA interaction network, as presented in Appendix A. This network was constructed using ENCODE ChIp-seq data in Networkanalyst 3.0 and generated through Cytoscape (version 3.10.1). Given that each hub gene was up-regulated in the unfavorable cluster 1/2 compared to the relatively good prognosis cluster 3, we submitted the list of hub genes to the cMAP website to conduct potential pharmacological analysis. The compounds were evaluated based on their connectivity scores, resulting in the identification of the top 20 potential compounds which are listed in Table 3, encompassing a diverse range of drug classes.

### 2.5. DRG-Related Risk Score Was Associated with Tumor Microenvironment (TME) Signature in DLBCL

The tumor microenvironment (TME) plays a crucial role in tumor progression and response to treatment. To explore the relationship between TME signature and DRG-related clusters in DLBCL, the ssGSEA infiltration algorithm was employed to evaluate immune cell abundance in individual samples (Figure 7A). The variation in immune cell population among the three clusters was notably significant, except for immature dendritic cells (iDCs), neutrophils, and Th1 cells. Particularly noteworthy was the significantly higher expression of gamma-delta T cells (Tgd) observed in the lower risk cluster 3, contrasting with the expression levels of CD8 T cells or DC cells. In Figure 7B, the relationship between the risk score and the signatures of activated DC cells, CD8 T cells, and Tgd was examined using Pearson’s correlation analysis. The expression of *CDE1L* in activated CD8 T cells showed a positive correlation with the risk score, whereas *TIMM13* exhibited a negative correlation. *RPS7* expression in activated Tgd was positively correlated with the risk score, while *KRT80* showed a negative correlation. Additionally, the expression of ATP6V1A in activated DC cells displayed a positive correlation with the risk score, whereas *BCL2L1* showed a negative correlation. Furthermore, the ESTIMATE result depicted in Figure 7C revealed that patients in cluster 2 exhibited the highest immune score and stromal score but the lowest TumorPurity among the three clusters.

## 3. Discussion

DLBCL is an aggressive B-cell non-Hodgkin lymphoma that affects patients of all ages with a range of clinical presentations, including painless swelling in the neck, armpit, or groin caused by enlarged lymph nodes, which can grow rapidly over a few weeks. DLBCL can also affect other parts of the body outside of lymph nodes (extranodal disease), leading to symptoms like abdominal discomfort or pain, diarrhea, bleeding, and the swelling of organs like the liver or spleen [20,21]. DLBCL itself encompasses a heterogeneous array of biologically distinct entities, culminating in the clonal proliferation of malignant B-cells derived from germinal or post-germinal origins. The diagnosis of this aggressive non-Hodgkin lymphoma typically involves the biopsy of suspicious lymph nodes or extranodal tumors, where the normal tissue architecture is replaced by sheets of large cells exhibiting positive staining for pan-B-cell antigens, like CD20 and CD79a [2]. In the past three decades, the International Prognostic Index has been employed to predict prognosis in aggressive non-Hodgkin lymphoma cases undergoing treatment with doxorubicin-containing regimens. Its validity has been confirmed in the rituximab era, revealing that patients with scores ranging from 0–1, 2, 3, 4–5 demonstrated respective 3-year overall survival rates of 91%, 81%, 65%, and 59% [22].

The diversity within DLBCL directly influences the significant variability in clinical responses to standard therapies, highlighting the need to explore innovative treatment approaches. There is significant evidence supporting the potential of therapies aimed at addressing disulfidptosis to serve as a novel strategy for cancer treatment [23,24,25,26]. Disulfidptosis stems from an imbalance in disulfide metabolism, a process involving the formation and rearrangement of disulfide bonds [27,28]. This intricate process is crucial for upholding protein stability [29,30,31]. Disulfide oxidoreductases are key players in this mechanism, facilitating the reduction and oxidation of disulfide bonds. These enzymes play a crucial role in preserving intracellular redox balance by promoting thiol–disulfide exchange [32,33]. In recent years, a number of studies have pointed to the fact that tumor cells face oxidative stress [16], which can disrupt disulfide metabolism [12], consequently affecting tumor survival and proliferation [23,34,35]. The dysregulation of disulfide metabolism is closely intertwined with redox balance, a critical mechanism in tumor development and progression. A significant proportion of tumors modulate the intracellular redox environment to enhance their tolerance, support survival, or facilitate metastasis [36,37,38]. For example, specific anti-tumor medications such as cisplatin and paclitaxel exert their anti-tumor effects by engaging with intracellular disulfide bonds [39,40,41]. Moreover, a considerable body of research has confirmed the significant role of SLC7A11-mediated ferroptosis in various diseases, including tumors [42,43,44,45]. Consequently, the balance between ferroptosis and disulfidptosis could emerge as a novel therapeutic approach to enhance the efficacy and survival rates of DLBCL patients.

This study marks the first comprehensive analysis of DRGs in DLBCL, highlighting the close link between disulfidptosis and DLBCL development. We identified eight DRGs to formulate a scoring model, which we then employed to assess 470 patients from the GSE31312 dataset. All patients were then categorized using an unsupervised clustering algorithm, resulting in the identification of three distinct disulfidptosis-related DLBCL subgroups. Interestingly, although cluster 3 displayed the lowest immune score level, it demonstrated the most favorable prognosis among the three subgroups. The immune infiltration analysis revealed that cluster 3 exhibited reduced infiltration of CD8 T cells and DC cells but increased infiltration of gamma-delta T cells, which correlated with its favorable prognosis. Following this, we examined the gene signatures of activated CD8+ T cells, DC cells, and gamma-delta T cells. Among all gene signature molecules, the expression level of CSE1L in activated CD8+ T cells showed the strongest positive correlation with risk score (cor = 0.56, *p* < 0.0001). CSE1L is a protein with a nuclear localization signal (NLS), and gene ontology annotations linked to it include binding and export signal receptor activities. Previous research has shown increased expression of CSE1L in bone marrow granulocytes of patients with primary chronic myeloid leukemia (CML). In the CML cell line K562 cells, CSE1L knockdown blocked the transition of the cell cycle from G0/G1 phase to S phase and enhanced apoptosis [46].

In this study, the subgroup characterized by high infiltration of gamma-delta T cells exhibited the most favorable prognosis. Gamma-delta T cells are the predominant component of intraepithelial lymphocytes in mucosal tissues [47]. Gamma-delta T cells comprise various highly adaptable subgroups and are relatively limited in the peripheral blood system [48,49]. The immune functionality of gamma-delta T cells displays diverse characteristics depending on the specific cell subtype and the surrounding microenvironment. For example, gamma-delta T17 cells, predominantly secreting IL-17, can exhibit two distinct properties: immune promotion or immune suppression [50,51,52]. The examination of 25 cancers, including CML, revealed that the genetic signature of tumor-associated gamma-delta T cells exhibited the most robust correlation with patient overall survival. Furthermore, elevated levels of gamma-delta T cells correlated positively with lower clinical stage and enhanced overall survival outcomes [53]. Additionally, immune-related inhibitory molecules and cells can attenuate the anti-tumor effect of gamma-delta T cells. For example, myeloid-derived suppressor cells, mesenchymal cells, M2 tumor-associated macrophages, and Treg cells within the tumor microenvironment can impede the activation of cytotoxic gamma-delta T cells by releasing immunosuppressive molecules [54,55,56]. The findings from these studies might support the idea that cluster 3 emerged as the most favorable prognostic subgroup in this study, characterized by higher infiltration of gamma-delta T cells and lower infiltration of Treg cells and macrophages. This underscores the beneficial role of gamma-delta T cells in DLBCL.

Of the eight genes identified for constructing the scoring system, *NDUFA11* stood out as the only discrepancy between cluster 1 and cluster 2, which exhibited the poorest prognosis. Notably, *NDUFA11* displayed the highest resemblance to other genes in the boxplot diagram illustrating similarity with the remaining seven genes, underscoring its pivotal role in the context of DLBCL. *NDUFA11* encodes a subunit of the mitochondrial respiratory chain NAD(P)H dehydrogenase complex I, the primary enzyme in the mitochondrial respiratory chain. Comprising 45 subunits, this complex transfers electrons from NAD(P)H to the respiratory chain. Its electron acceptor is ubiquinone, and the enzyme serves as the primary scavenger of ROS, aiding in their neutralization [57]. *NDUFA11* mutation has been linked to destabilizing complex I, resulting in infantile lactic acidemia or cerebral cardiomyopathy [58]. Other related conditions of *NDUFA11* includes late-onset myopathy [59], nuclear type 14, and isolated complex I deficiency.

The *p*-value associated with *CAPZB* in the multivariate regression analysis was determined to be less than 0.05, indicating its potential as an independent prognostic factor (*p* = 0.007). *CAPZB*, belonging to the F-actin capping protein family, encodes the beta subunit of actin-binding proteins. It holds a crucial role in regulating the thickness of the mitotic cortex. Changes in cortex thickness influence mitotic tension, consequently affecting cell cycle progression. Specifically, variations in actin cortex thickness and tension directly influence cell surface tension, thus modulating cell cycle dynamics [60]. This observation provides a feasible rationale for the lowering expression of *CAPZB* in cluster 3. Considering its involvement in regulating mitotic cortex thickness and its impact on cell cycle progression, decreased expression of *CAPZB* might contribute to the distinct characteristics observed in cluster 3, including favorable prognosis and higher infiltration of gamma-delta T cells.

In contrast to *CAPZB*, *MYH9* encodes the myosin IIA heavy chain and may serve as an independent protective prognostic factor (*p* = 0.001). MYH9 contains an IQ domain and a myosin head-like domain, playing a crucial role in various essential functions such as cytokinesis, cell motility, and cell shape maintenance. Previous studies have suggested that during critical steps in cell invasion, spreading, and migration, MYH10 facilitates lamellipodia extension by positioning its motor at the leading-edge during cell spreading. In contrast, MYH9 is positioned adjacent to MYH10 to weaken or contract lamellipodia, leading to a halt in cell spreading [61]. In AML, disruption of myosin contractility has been shown to enhance AML apoptosis [62], a finding consistent with the results observed in this study.

DSTN, known as destrin, which functions as an actin-depolymerizing protein, is involved in severing actin filament (F-actin) and binding to actin monomers (G-actin) in a pH-independent manner [63]. This protein plays a crucial role in actin filament turnover and cytoskeletal reorganization, thereby modulating cell motility and cytokinesis [64].

NDUFS1, similar to NDUFA11, functions as a subunit of complex I (NADH: ubiquinone oxidoreductase), playing an essential role in the assembly and stability of complex I. Additionally, NDUGS1 participates in the interaction between complex I and the ubiquinol–cytochrome reductase complex (complex III), facilitating the formation of a super-complex [65,66].

IQGAP1, abbreviated for IQ motif-containing GTPase-activating protein 1, interacts with cytoskeletal elements, cell adhesion molecules, and various signaling factors to regulate cellular morphology and motility. It has been demonstrated to play a regulatory role by promoting cell cycle progression following DNA replication arrest [67]. Additionally, overexpression of wild-type IQGAP1 has been found to facilitate neurite outgrowth in neuroblastoma cells [68].

The protein encoded by *GYS1* facilitates the addition of glucose monomers to generate glycogen molecules through the creation of α-1,4-glycosidic bonds. Its associated pathways include cAMP-dependent activation of PKA and glycogen metabolism. Research indicates that the elevated expression of GYS1 is linked to unfavorable prognosis in acute myeloid leukemia (AML) and myelodysplastic syndrome (MDS) [69].

*OXSM* represents the final potential independent prognostic factor, with limited studies conducted on this gene. Previous research indicates that *OXSM* encodes beta-ketoacyl synthase, which likely participates in the biosynthesis of lipoic acid and long-chain fatty acids essential for optimal mitochondrial function [70].

Analyzing the differential genes across various subgroups, those up-regulated in the subgroup with poor prognosis predominantly encompass functions related to myeloid cell differentiation (GO:0030099), cadherin binding (GO:0045296), and heat shock protein binding (GO:0031072). Additionally, pathways associated with DNA replication and cell cycle regulation are notably enriched in cluster 1/2 compared to cluster 3 with a favorable prognosis. Cadherins represent homophilic adhesion molecules crucial for mediating cell–cell adhesion and regulating tissue morphogenesis. Additionally, the up-regulation of cadherin-binding pathways appears to be intricately associated with multiple mechanisms that facilitate cancer cell spread and metastasis. During epithelial–mesenchymal transition (EMT), a process frequently observed in cancer progression, cancer cells commonly down-regulate the expression of E-cadherin, while concurrently up-regulating other cadherins such as N-cadherin and P-cadherin [71,72,73]. This phenomenon of “cadherin switching” enhances the migratory and invasive capabilities of cancer cells, contributing to their ability to metastasize and spread throughout the body [73]. The up-regulation of P-cadherin and N-cadherin can trigger the activation of signaling pathways associated with cancer cell growth, migration, and invasion, including receptor tyrosine kinase, PI3K/AKT, Rho GTPases, and Hippo pathways [71,72]. This suggests that cells within the poor prognosis subgroup likely possess enhanced abilities for cell migration, invasion, and scattering when compared to those within the subgroup with a better prognosis.

The compounds identified through our screening are based on the DEGs from the subtypes with poor prognosis compared with the subgroup with better prognosis. The list of compounds encompasses a diverse array of types. According to research findings, HMGCR inhibitors, commonly known as statins, have demonstrated promising anti-cancer effects, including efficacy against leukemia [74,75,76]. Studies have shown that deregulated or heightened activity of HMG-CoA reductase is observed in various cancers, including leukemia. In vitro investigations have revealed that statins can induce apoptosis in acute myelogenous leukemia in a sensitive and specific manner [74,75]. However, Cerivastatin, which ranked highest in the current analysis, was withdrawn from the market due to causing rhabdomyolysis both when administered alone and in combination with gemfibrozil [77,78,79]. At present, the practical choice is to use other statins with milder side effects as alternatives.

Palbociclib is a CDK4/6 inhibitor that has shown promise in the treatment of specific types of leukemia. Palbociclib is approved for use in combination with other medications to treat hormone receptor-positive (HR+), HER2-negative advanced, or metastatic breast cancer [80]. Preclinical studies have shown that palbociclib enhances the apoptotic effect induced by doxorubicin in DLBCL, especially ABC type [81].

Tacedinaline, also known as CI-994, is a class I histone deacetylase (HDAC) inhibitor, which has been found to induce apoptosis, and growth arrest in preclinical models of AML and non-small cell lung cancer cell lines [82,83].

Valdecoxib, a selective inhibitor of the cyclooxygenase-2 enzyme and classified as a non-steroidal anti-inflammatory drug (NSAID), was previously used in the treatment of arthritis, rheumatoid arthritis, and pain associated with menstrual periods [84]. However, it was voluntarily withdrawn from the U.S. market in 2005 due to safety concerns, including an elevated risk of serious cardiovascular events and potentially life-threatening skin reactions.

Several studies have shown that inhibiting ATPase subunits of the 19S proteasome (a part of the 26S proteasome complex) can induce apoptosis in AML [85,86,87]. mTOR is a protein kinase that regulates cellular process including cell growth, survival, and metabolism. Dysregulation of mTOR has been implicated in the development of various types of cancers [88,89]. Several classes of mTOR inhibitors such as rapalogues or second-generation mTOR inhibitors have been evaluated in clinical trials for leukemia treatment [90].

Salt-inducible kinase 3 (SIK3) has been emerged as a potential therapeutic target in AML [91]. The studies show that the inhibition of SIK3 in vitro and in vivo can suppress the function of the transcription factor MEF2C and inhibit the proliferation of AML cells, either with small molecule inhibitors or genetically [91]. This evidence suggests the potential therapeutic functions of a sodium/potassium/chloride transporter inhibitor.

It is noteworthy that among other categories of small molecules, Hsp90 inhibitors were found to be an effective candidate against adult T-cell leukemia (ATL), and AML cells. In AML cells, Hsp90 inhibitor PU-H71 induced cell cycle arrest, protein degradation, and apoptosis, especially in treatment combined with BH3-mimetic drugs that target anti-apoptotic proteins [92]. Additionally, the category including 17-DMAG, NVP-AUY922, and TAS-116 were demonstrated to have significant suppressive activity against ATL [93,94]. The detailed clinical trials and the research/development (Candidate 5) procession of each small molecular candidate and the target are shown in Appendix A.

In this study, we examined the expression and potential impact of disulfide death-related genes in DLBCL using data sets GSE31312 and GSE12453. By employing a combined machine learning algorithm and univariate–Cox LASSO multivariate–Cox regression, we developed a prognostic model and stratified 470 patients into three subgroups, revealing significant prognostic differences among them. This highlights the potential influence of disulfidptosis in DLBCL. Additionally, we analyzed the pathways affecting prognosis based on differential genes between subgroups and identified candidate small molecule drugs capable of extending the survival of DLBCL patients.

Disulfidptosis has emerged as a newly discovered form of programmed cell death, attracting growing attention in tumor research and related diseases. Our groundbreaking findings highlight the potential of leveraging disulfidptosis as a treatment modality for DLBCL. In recent years, various novel treatment modalities, particularly CAR-T therapy [95] and bispecific antibodies [96], have been approved or are being attempted in clinical trials for relapsed/refractory DLBCL. The use of disulfidptosis-inducing agents that we proposed as a complementary mechanism to these treatments, or the application of antibody–drug conjugates as drugs [97], is also sufficiently conceivable.

However, this article also acknowledges its limitations. For instance, while the LASSO algorithm serves as a compressed estimation technology, its inability to obtain an explicit solution, unstable results, and potential errors in approximate calculations presents challenges. To address this question, we employed the random forest algorithm to analyze both 24 DRGs and the subset of 8 DRGs identified by the predictive model (Appendix A). The random forest approach incorporated all eight DRGs within its feature factors, achieving an accuracy of 65% through 10-fold cross-validation. We observed that the hierarchy of feature importance derived from the random forest algorithm diverged from that obtained via the FRIEND algorithm. This divergence can be attributed to the differing methodologies employed by the two approaches: while the FRIEND algorithm evaluates feature importance through the analysis of interactions among features within an ensemble learning framework, the random forest algorithm constructs decision trees based on randomly selected subsets of features, focusing primarily on the individual contribution of each feature to the prediction outcome. Interestingly, when evaluating the relative importance of the complete set of 24 DRGs, OXSM was not featured in the variable importance list. This may be contributed by the variable sensitivities of these algorithms to details within the data. The continual development and refinement of such algorithms are essential to deepen our understanding and enhance the robustness of research findings in this field. Therefore, further research and timely updates to the algorithm are warranted. In addition, the specific mechanism of disulfidptosis requires further study to confirm the current findings, and the predictive model established in this study also needs to be validated in practice.

## 4. Materials and Methods

### 4.1. Data Acquisition and Preprocessing

In this study, our investigation commenced with the retrieval of the keyword “DLBCL” from the PubMed and Gene Expression Omnibus (GEO) databases. This search led to the identification of two significant gene expression profiles: GSE31312 [98,99], kindly contributed by C. Visco, Y. Li, Z. Y. Xu-Monette, and R. N. Miranda, and GSE12453 [100], generously provided by V. Brune, E. Tiacci, I. Pfeil, and C. Döring et al. Both datasets were generated utilizing the GPL570 (HG-U133_Plus_2) Affymetrix Human Genome U133 Plus 2.0 Array platform, marking a pivotal foundation for our research. GSE31312 comprises data from 498 de novo adult DLBCL cases; the RNA samples from DLBCL patients underwent extraction and amplification through Single Primer Isothermal Amplification (SPIA, NuGen Inc, San Carlos, CA, USA). Following this, the amplified RNA samples were then hybridized to Affymetrix HG-U133 Plus 2.0 GeneChips, facilitating the gene expression profiles associated with DLBCL. Patients lacking clinical information were excluded from the study, resulting in the data from 470 patients being incorporated into the final analysis. GSE12453 generated 36 gene expression profiles, comprising classical Hodgkin lymphoma cells, non-Hodgkin lymphoma B cell, T-cell–rich B-cell lymphoma and five normal B-cell subsets. Within this collection, this study specifically included 11 DLBCL samples, 5 naive B-cell samples, 5 memory B-cell samples, 5 centrocyte samples, and 5 centroblast samples, to evaluate the different expression of disulfidptosis-related genes in DLBCL compared with normal B cells. The raw CEL files of these two datasets were downloaded from the GEO database and raw array data were uniformly preprocessed and normalized, respectively, using the robust multi-array average (RMA) algorithm and normalized BetweenArrays from R package ‘limma’ (version no. 3.58.1).

### 4.2. Collecting Disulfidptosis-Related Genes (DRGs) via Systematic Review

The 24 DRGs in the present investigation were identified from a previous study in the PubMed database (Table 1). Detailed information for the DRGs was obtained from GeneCards [101] (https://www.genecards.org/, accessed on 1 February 2024) and The Human Protein Atlas (HPA, https://www.proteinatlas.org/, accessed on 1 February 2024). The gene ontology (GO) functional enrichment analysis [102] and Kyoto Encyclopedia of Genes and Genome (KEGG) pathway analysis [103] specifically for DRGs were carried out by Metascape [104] (https://metascape.org/gp/index.html#/main/step1, accessed on 1 February 2024) and ggplot2 (version 3.3.6) package in R, with results visualized through the Xiantao website. The *p* value cutoff of inclusion criterion for statistical significance was chosen as 0.05.

### 4.3. Screening Prognosis-Associated DRGs in DLBCL

In order to determinate the prognostic DRGs in DLBCL, the univariate Cox regression, LASSO regression, and multivariable Cox regression analysis were utilized to construct the novel prognostic gene signature, by “survival (version 3.5.7)” “glmnet (version 4.1.8)” R (version 1.0.13) package, with a cutoff value as *p* < 0.05. Finally, eight DRGs, including CAPZB, DSTN, GYS1, IQGAP1, MYH9, NDUFA11, NDUFS1, and OXSM were selected to establish the prognosis-associated DRGs to establish a prognostic model for patients with DLBCL. All 470 participated cohorts were divided into low-risk and high-risk groups based on the risk score with median cut-off value. The Kaplan–Meier (K-M) survival curve and ROC curve were performed by “survival”, “survminer (version 0.4.9)”, and “timeROC (0.4)” R package was used to assess the survival conditions and predictive performance for the prognosis signature, respectively. The FRIENDS analysis was performed to account for the average similarity of each gene to others in eight prognosis-associated DRGs by GOSemSim (version no 2.22.0). Each sample’s risk score was calculated using the following formula:$$\sum_{k=1}^{i}expression\;gene\;i\times coefi$$ Random forest was implemented in R with “caret” package (version No. 6.0.94) with leave-one-out 10-fold cross validation.

### 4.4. Unsupervised Clustering for Eight Prognosis-Associated DRGs

Based on eight prognosis-associated DRGs, unsupervised classification was performed using a hierarchical consensus approach by R package “ConsensuClusterPlus” with 1000 repetitions. The prognosis-related DRGs were separated into three clusters. A survival curve was used to study the survivability of eight DRGs and the limma package was utilized to evaluate the changes in particular genes between subtypes.

### 4.5. Cluster Differentially Expressed Genes (DEGs) and Enrichment Analysis

To assess a substantial change of DEGs between three clusters (cluster1 vs. cluster3, cluster2 vs. cluster3), |log2FC|≥1 and *p*.adj < 0.05 were considered as significant. GSEA analysis was performed to evaluate the median gene expression level of cluster 1/2 compared with cluster 3 to explore the different biological signaling routes. The protein–protein interaction network and hub genes were constructed by STRING [105,106] (version 12.0) combined with Cytoscape [106] (version 3.10.1) plug-in Cytohubba employing the Maximal Clique Centrality algorithm. Gene ratio, *p*-value, and Z-score were used to identify the GO/KEGG pathway with significant enrichment data linked with the DEGs. BP, MF, and CC regulated by DEGs were explored by the R package “clusterProfiler (version 4.4.4)”, “Goplot (version 1.0.2)” and visualized through the Xiantao website.

### 4.6. Potential Pharmacological

The up-regulated DEGs of cluster 1/2 compared with cluster 3 were uploaded to the ConnectivityMap [107] (cMap, https://clue.io/, accessed on 1 February 2024) database. The small molecular compounds with a negative score of connectivity were exported, which may trigger gene expression changes in an opposite direction and hold a function of extended lifespan for the patients with DLBCL.

### 4.7. Tumor Microenvironment and Immune Cell Infiltration

The StromalScore, ImmuneScore, ESTIMATEScore, and TumorPurity were calculated using the “estimate (version 1.0.13)” package by R [108]. The enriched values of 24 immune cells in all three cohorts were investigated through the single sample GSEA (ssGSEA) algorithm based on the R package “GSVA (version 1.44.5)”. The ssGSEA infiltration algorithm uses markers specific to each type of immune cell as a gene set to calculate the enrichment score of each type of immune cell in each sample and infer individual immune cell infiltration of the sample [109,110]. Gene signatures of each tumor-infiltrating lymphocytes was downloaded from the TISIDB [111] website (http://cis.hku.hk/TISIDB/download.php, accessed on 1 February 2024).

### 4.8. Statistical Analysis

R (version 4.3.2) and the GraphPad Prism (version 9.4.1) were applied for statistical analysis. A multiple unpaired *t* test was used to analyze the expression level of DRGs in DLBCL compared with each kind of normal B cell. StromalScore, ImmuneScore, ESTIMATEScore, and TumorPurity in each cluster was performed by one-way ANOVA with Tukey’s multiple comparison test. For statistical analysis of immune infiltration in each cluster, a two-way ANOVA with Tukey’s multiple comparison test was performed. Statistical significance was analyzed using the one-way analysis of variance (ANOVA) and two-way ANOVA test with Tukey’s multiple comparison tests for StromalScore, (ImmuneScore, ESTIMATEScore, TumorPurity) and immune infiltration, respectively. The correlation analysis was evaluated using the Pearson method.

## Figures and Tables

**Figure 1 ijms-25-07156-f001:**
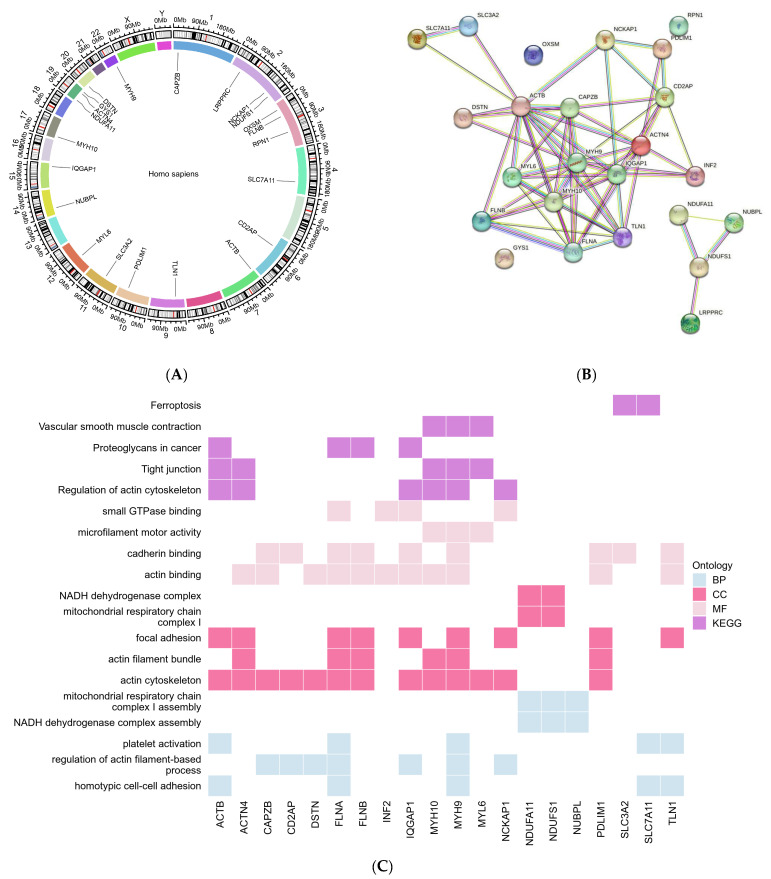
**Landscape of disulfidptosis**-**related genes (DRGs) in DLBCL.** (**A**) The chromosomal localization of DRGs. (**B**) Protein–protein interactions among 24 DRGs. (**C**) Heatmap visualization of GO/KEGG analysis illustrating the function and pathways associated with each DRG. (**D**) Spearman correlation comparing the expression of 24 DRGs in DLBCL (**left**) and normal B cells (**right**). The four scatter-plots highlight the most correlated DRGs in DLBCL and normal B cells, respectively.

**Figure 2 ijms-25-07156-f002:**
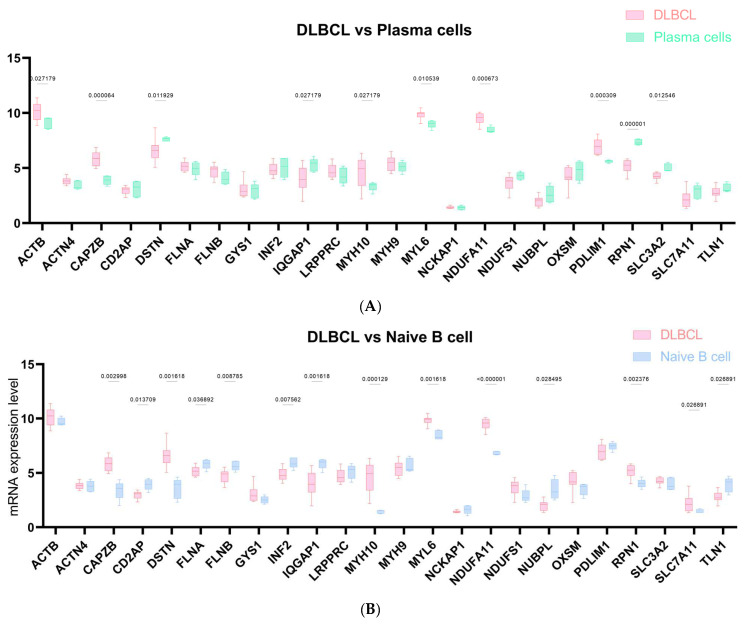
**The transcriptome expression profiles of 24 DRGs were compared between DLBCL and the following cell types:** (**A**) Plasmas cells, (**B**) naive B cells, (**C**) memory B cells, (**D**) centrocytes, and (**E**) centroblasts. (**F**) Heatmap visualization of 24 DRGs’ expression level of DLBCL and five normal subtypes of B cells.

**Figure 3 ijms-25-07156-f003:**
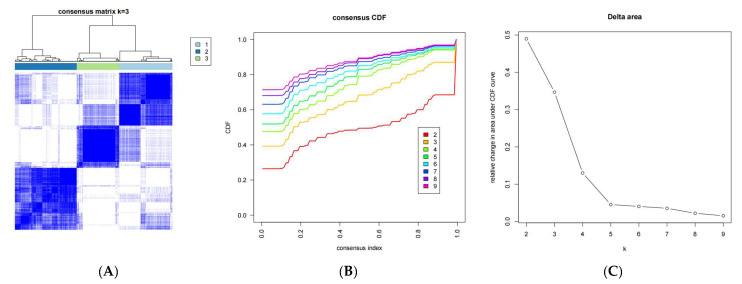
**The prognostic prediction of DLBCL influenced by DRGs, as evidenced by** (**A**) unsupervised clustering of 24 DRGs into three clusters, (**B**) consensus clustering cumulative distribution function (CDF) for k = 2–9, (**C**) relative change in area under the CDF curve for k = 2–9, (**D**) principal component analysis (PCA) for the transcriptome profiles of three DRG subtypes, (**E**) heatmap illustrating the expression levels of 24 DRGs in three distinct DRG subtypes, and (**F**) the Kaplan–Meier curve demonstrating a difference in overall survival (OS) among three DRG clusters.

**Figure 4 ijms-25-07156-f004:**
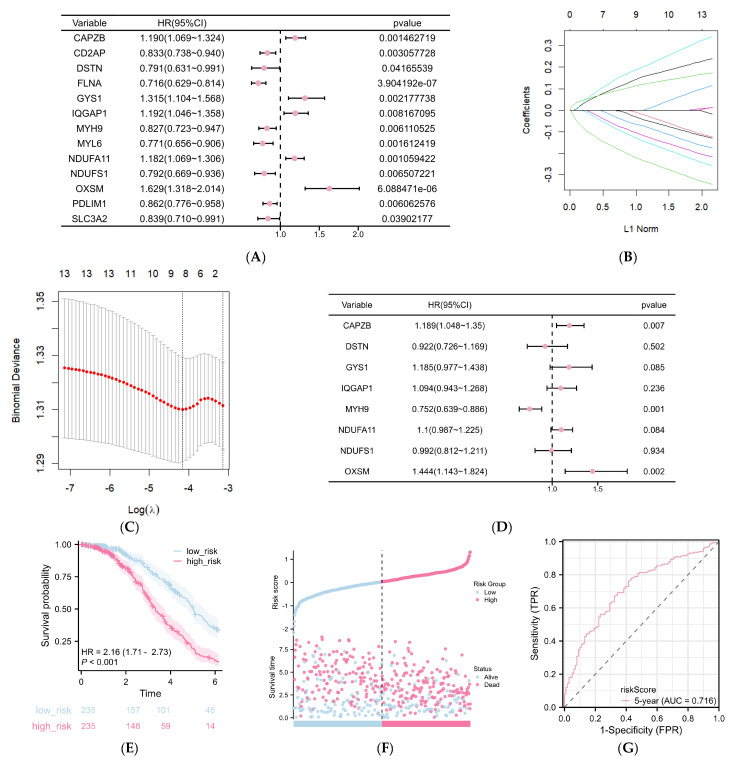
Risk signature with eight DRGs. (**A**) Forest plots of univariate Cox regression analysis based on 24 DRGs. (**B**) Ten-fold cross-validation for tuning parameter selection in LASSO model. (**C**) LASSO coefficient profiles of 13 DRGs. (**D**) Forest plots of multivariate Cox regression analysis of eight DRGs. (**E**) Survival analysis for high-risk and low-risk groups. (**F**) Risk score and survival outcome of each sample in GSE31315. (**G**) The five-year ROC of the model is 0.716.

**Figure 5 ijms-25-07156-f005:**
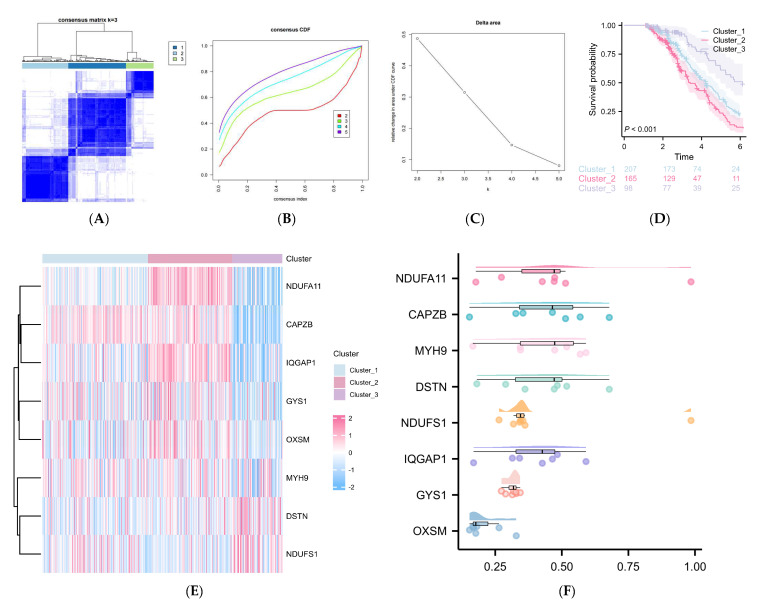
The cluster variance analysis according to eight DRGs. (**A**) Unsupervised clustering of eight DRGs into three clusters, cluster 1 (44%), cluster 2 (35%), cluster 3 (21%). (**B**) Consensus clustering cumulative distribution function for k = 2–9. (**C**) Relative change in the area under CDF curve for k = 2–9. (**D**) Kaplan–Meier curve showing different overall survival (OS) among three DRG clusters. (**E**) Heatmap illustrating the expression status of eight DRGs in three clusters. (**F**) FRIEND analysis of eight DRGs, displaying the descending order ranking based on the similarity of each DRG.

**Figure 6 ijms-25-07156-f006:**
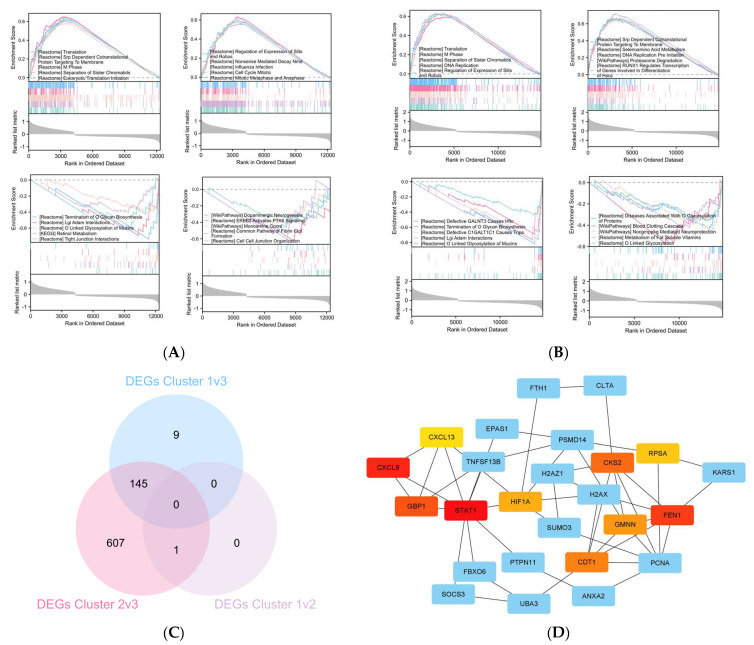
Landscape of DEGs in cluster 1 and 2 relative to cluster 3. (**A**) GSEA analysis of cluster 1. (**B**) GSEA analysis of cluster 2. (**C**) Venn diagram. (**D**) Protein–protein interactions among DEGs, the top ten hub genes were highlighted in gradient red, ranked by their MCC values.

**Figure 7 ijms-25-07156-f007:**
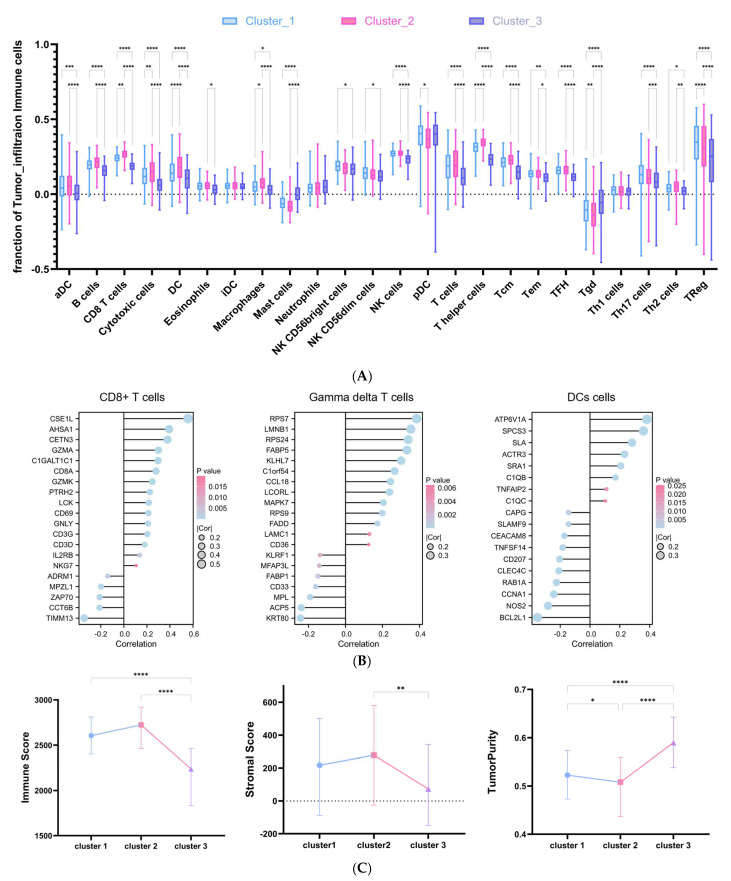
The different immune profiles of each cluster. (**A**) Enrichment values of 24 immune cells in all three clusters studied using the ssGSEA algorithm. (**B**) Correlation between risk score and the expressions of activated CD8+ T cells, gamma delta T-cells, and DC cells. (**C**) ESTIMATE algorithm analysis (* *p* < 0.1, ** *p* < 0.05, *** *p* < 0.001, **** *p* < 0.0001).

**Table 1 ijms-25-07156-t001:** Twenty-four disulfidptosis-related genes were compiled for this study.

Gene Symbol	Subcellular Location	Chromosomal Position
*ACTB*	intracellular	*chr7:5,526,409−5,563,902*(−)
*ACTN4*	Actin filaments/Cytosol	*chr19:38,647,649−38,731,589*(*+*)
*CAPZB*	Nucleoplasm/Cytosol/Vesicles	*chr1:19,338,775−19,485,539*(−)
*CD2AP*	Plasma membrane/Centriolar satellite	*chr6:47,477,789−47,627,263*(*+*)
*DSTN*	Plasma membrane	*chr20:17,570,075−17,609,919*(*+*)
*FLNA*	Plasma membrane /Actin filaments/Cytosol	*chrX:154,348,524−154,374,634*(−)
*FLNB*	Plasma membrane/Golgi apparatus/Actin filament/Cytosol	*chr3:58,008,398−58,172,251*(*+*)
*GYS1*	Microtubules/Cytosol	*chr19:48,968,130−48,993,310*(−)
*INF2*	Endoplasmic reticulum/Nuclear bodies	*chr14:104,681,146−104,722,535*(*+*)
*IQGAP1*	Plasma membrane/Cell Junctions	*chr15:90,388,242−90,502,239*(*+*)
*LRPPRC*	Mitochondria	*chr2:43,886,224−43,996,265*(−)
*MYH10*	Actin filaments/Mitochondria/Cytosol	*chr17:8,474,207−8,631,376*(−)
*MYH9*	Plasma membrane/Actin filaments/Nuclear bodies (uncertain)/Cytosol	*chr22:36,281,280−36,388,010*(−)
*MYL6*	cytosol/cytoskeleton/Supramolecular fiber	*chr12:56,158,346−56,163,496*(*+*)
*NCKAP1*	Cytosol	*chr2:182,909,115−183,038,858*(−)
*NDUFA11*	Mitochondria	*chr19:5,891,229−5,904,006*(−)
*NDUFS1*	Mitochondria	*chr2:206,114,817−206,159,509*(−)
*NUBPL*	Mitochondria	*chr14:31,489,956−31,861,224*(*+*)
*OXSM*	Mitochondria/Cytosol (uncertain)	*chr3:25,782,917−25,794,534*(*+*)
*PDLIM1*	Actin filaments Plasma membrane (uncertain)/Cell Junctions (uncertain)	*chr10:95,237,572−95,291,012*(−)
*RPN1*	Endoplasmic reticulum/Cytosol	*chr3:128,619,969−128,681,075*(−)
*SLC3A2*	Plasma membrane/Nucleoplasm (uncertain)	*chr11:62,856,004−62,888,880*(*+*)
*SLC7A11*	Vesicles	*chr4:138,164,097−138,312,671*(−)
*TLN1*	Focal adhesion sites/Cytosol/Plasma membrane/Centriolar	*chr9:35,696,948−35,732,195*(−)

**Table 2 ijms-25-07156-t002:** The detailed information list of eight prognostic DRGs.

Gene Symbol	Full Name	Location	Function of the Encoded Protein
*CAPZB*	Capping Actin Protein of Muscle Z-Line Subunit Beta	Nucleoplasm/ Cytosol/ Vesicles	F-actin-capping proteins bind in a Ca(2+)-independent manner to the fast-growing ends of actin filaments (barbed end) thereby blocking the exchange of subunits at these ends.
*DSTN*	Destrin, Actin Depolymerizing Factor	Plasma membrane	Actin-depolymerizing protein. Severs actin filaments (F-actin) and binds to actin monomers (G-actin).
*GYS1*	Glycogen Synthase 1	Microtubules/ Cytosol	Transfers the glycosyl residue from UDP-Glc to the non-reducing end of alpha-1,4-glucan.
*IQGAP1*	IQ Motif Containing GTPase Activating Protein 1	Plasma membrane/ Cell Junctions	Could serve as an assembly scaffold for the organization of a multimolecular complex that would interface incoming signals to the reorganization of the actin cytoskeleton at the plasma membrane.
*MYH9*	Myosin Heavy Chain 9	Plasma membrane/ Actin filaments/ Nuclear bodies (uncertain)/Cytosol	Cellular myosin that appears to play a role in cytokinesis, cell shape, and specialized functions such as secretion and capping.
*NDUFA11*	NADH:Ubiquinone Oxidoreductase Subunit A11	Mitochondria	Accessory subunit of the mitochondrial membrane respiratory chain NADH dehydrogenase (Complex I), that is believed not to be involved in catalysis.
*NDUFS1*	NADH:Ubiquinone Oxidoreductase Core Subunit S1	Mitochondria	Core subunit of the mitochondrial membrane respiratory chain NADH dehydrogenase (Complex I) which catalyzes electron transfer from NADH through the respiratory chain, using ubiquinone as an electron acceptor.
*OXSM*	3-Oxoacyl-ACP Synthase, Mitochondrial	Mitochondria/ Cytosol (uncertain)	May play a role in the biosynthesis of lipoic acid as well as longer chain fatty acids required for optimal mitochondrial function.

**Table 3 ijms-25-07156-t003:** Top twenty potential compounds were screened based on DEGs of cluster 1/2 compared with better prognosis cluster 3.

Score	ID	Name	Description
−99.89	BRD-K81169441	cerivastatin	HMGCR inhibitor
−99.82	BRD-K51313569	palbociclib	CDK inhibitor
−99.58	BRD-K52313696	tacedinaline	HDAC inhibitor
−99.58	BRD-K12994359	valdecoxib	Cyclooxygenase inhibitor
−99.51	BRD-K05350981	oligomycin-c	ATPase inhibitor
−99.47	BRD-A45498368	WYE-125132	MTOR inhibitor
−99.47	BRD-K63630713	etacrynic-acid	Sodium/potassium/chloride transporter inhibitor
−99.37	BRD-K65503129	HSP90-inhibitor	HSP inhibitor
−99.37	BRD-K53523901	arctigenin	MEK inhibitor
−99.33	BRD-K08417745	SID-26681509	Cathepsin inhibitor
−99.3	BRD-A48261811	argatroban	Thrombin inhibitor
−99.3	BRD-K56429665	calcipotriol	Vitamin D receptor agonist
−99.19	BRD-K51805276	temefos	Cholinesterase inhibitor
−99.19	BRD-K32828673	chelidonine	Tubulin inhibitor
−99.15	BRD-A72703248	SKF-96365	Calcium channel blocker
−99.08	BRD-K90382497	GW-843682X	PLK inhibitor
−99.01	BRD-K93201660	ML-7	Myosin light chain kinase inhibitor
−99.01	BRD-A11678676	wortmannin	PI3K inhibitor
−98.98	BRD-K13049116	BMS-754807	IGF-1 inhibitor

## Data Availability

GSE31312 and GSE12453 datasets were download from the Home-GEO-NCBI https://www.ncbi.nlm.nih.gov/geo/.

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
