# Peer review of "Disulfidptosis: A Novel Prognostic Criterion and Potential Treatment Strategy for Diffuse Large B-Cell Lymphoma (DLBCL)"

_ijms, 2024, doi:10.3390/ijms25137156_

Round 1

Reviewer 1 Report

Comments and Suggestions for Authors

The current manuscript entitled “Disulfidptosis: A Novel Prognostic Criterion and Potential Treatment Strategy for Diffuse Large B-cell Lymphoma (DLBCL)” by Wang and college investigated the expression variations of disulfidptosis-related genes (DRGs) in DLBCL using two publicly available gene expression datasets. The author shows difference in gene expression pattern in normal B cells vs DLBCL. Moreover, the authors also revealed that DRGs strongly associated with prognostic outcomes, revealing 8 characteristic DRGs (CAPZB, DSTN, GYS1, IQGAP1, MYH9, NDUFA11, NDUFS1, OXSM). All together the authors conclude that DRGs can predict prognosis and can help identify novel therapeutic candidates. The authors have done a great job in clearly addressing the results and discussion. The manuscript is well written. Therefore,  I strongly recommend this article to be published in this journal.

Author Response

To Reviewer 1

We sincerely appreciate your high evaluation. Reviewer 2 has raised several questions, and we have updated our response in version 2 to address these concerns. We would be grateful if you could review it.

Reviewer 2 Report

Comments and Suggestions for Authors

This study is the first to analyze disulfidptosis-related genes (DRGs) in DLBCL, identifying eight key DRGs and creating a prognostic scoring model. Analysis of 470 patients from the GSE31312 dataset categorized them into three subgroups, with the lowest immune score subgroup showing the best prognosis. High gamma-delta T cell infiltration was associated with improved outcomes. Among the DRGs, NDUFA11 was notably significant. Potential therapeutic compounds, such as statins and CDK4/6 inhibitors, were identified. Despite promising findings, limitations like LASSO algorithm instability necessitate further research.

Authors mentioned unclassified DLBCL subtype. However, the frequency among DLBCL patients was not mentioned.

Authors may complement LASSO with other feature selection methods like Ridge Regression, Elastic Net, or machine learning techniques (e.g., Random Forest, SVM) to compare results and enhance stability.

Investigate the limitations of the LASSO algorithm by using ensemble methods or cross-validation techniques to ensure the robustness of the predictive model.

I wonder if authors did same analysis in other lymphomas to make a comparison. 

Author Response

To Reviewer 2

Thank you for your insightful comments and suggestions. We have added information regarding the incidence rates of DLBCL subtypes in the legend of Figure 5. The frequencies of each subtype are as follows: cluster1 at 44%, cluster2 at 35%, and cluster3 at 21%.

Regarding your concerns about the limitations of the LASSO algorithm, we are in full agreement. The LASSO algorithm is generally considered to have the drawback of limiting the number of predictive factors selected based on the sample size. Therefore, we have utilized a large dataset of 470 samples to mitigate the limitations these limitations.

Unlike Ridge and Elastic Net regression, LASSO has the unique characteristic of selecting one or a few predictors from a subset of correlated predictors, shrinking the coefficients of the remaining predictors to zero. In contrast, Ridge regression only shrinks the coefficients of all features without reducing any to zero. Our initial intention was to narrow down the range of predictors and identify the most suitable ones, which is why we decided to forgo Ridge and Elastic Net regression.

In response to your suggestion, we have applied the random forest algorithm to both the 24-DRGs and the 8-DRGs selected by our model. The results have been added to Supplementary Figure 6 and discussed in the final section of the Discussion. All changes are highlighted in blue.

When using the LASSO model to select features, we employed 10-fold cross-validation to optimize the regularization parameter, as indicated in blue in the legend of Figure 4. The parameters were set as follows:

cv_fit = cv.glmnet(x, y, family = "binomial", alpha = 1, type.measure = "deviance", nfolds = 10)

Regarding the last point, due to the limited availability of datasets that meet the criteria, we have not performed similar analyses on other types of lymphoma.

We once again thank you for your valuable suggestions.